# Impact of onset-to-door time on outcomes and factors associated with late hospital arrival in patients with acute ischemic stroke

**Eung-Joon Lee**[1‡], **Seung Jae Kim**[2,3‡], **Jeonghoon Bae**[1], **Eun Ji Lee**[4], **Oh Deog Kwon**[5], **Han-Yeong Jeong**[1], **Yongsung Kim**[6], **Hae-Bong Jeong**[6]*

1 Department of Neurology, Seoul National University Hospital, Seoul, Republic of Korea, 2 Department of Family Medicine, Seoul St. Mary's Hospital, College of Medicine, The Catholic University of Korea, Seoul, Republic of Korea, 3 International Healthcare Center, Seoul St. Mary's Hospital, College of Medicine, The Catholic University of Korea, Seoul, Republic of Korea, 4 Department of Radiology, Soonchunhyang University Seoul Hospital, Seoul, Republic of Korea, 5 Republic of Korea Navy 2nd Fleet Medical Corps, Pyeongtaek-si, Gyeonggi-do, Republic of Korea, 6 Department of Neurology, Chung-Ang University Hospital, Seoul, Republic of Korea

‡ These authors contributed equally to this work as co-first authors on this work.
* hbjeong315@gmail.com

## Abstract

### Background and purpose

Previous studies have reported that early hospital arrival improves clinical outcomes in patients with acute ischemic stroke; however, whether early arrival is associated with favorable outcomes regardless of reperfusion therapy and the type of stroke onset time is unclear. Thus, we investigated the impact of onset-to-door time on outcomes and evaluated the predictors of pre-hospital delay after ischemic stroke.

### Methods

Consecutive acute ischemic stroke patients who arrived at the hospital within five days of onset from September 2019 to May 2020 were selected from the prospective stroke registries of Seoul National University Hospital and Chung-Ang University Hospital of Seoul, Korea. Patients were divided into early (onset-to-door time, ≤4.5 h) and late (>4.5 h) arrivers. Multivariate analyses were performed to assess the effect of early arrival on clinical outcomes and predictors of late arrival.

### Results

Among the 539 patients, 28.4% arrived early and 71.6% arrived late. Early hospital arrival was significantly associated with favorable outcomes (three-month modified Rankin Scale [mRS]: 0−2, adjusted odds ratio [aOR]: 2.03, 95% confidence interval: [CI] 1.04–3.96) regardless of various confounders, including receiving reperfusion therapy and type of stroke onset time. Furthermore, a lower initial National Institute of Health Stroke Scale (NIHSS) score (aOR: 0.94, 95% CI: 0.90–0.97), greater pre-stroke mRS score (aOR: 1.58, 95% CI: 1.18–2.13), female sex (aOR: 1.71, 95% CI: 1.14–2.58), unclear onset time, and ≤6 years of schooling (aOR: 1.76, 95% CI: 1.03–3.00 compared to >12 years of schooling) were independent predictors of late arrival.

**Data Availability Statement:** All relevant data are within the paper and its Supporting Information files.

**Funding:** This research was supported by a fund (#2020ER620200) by the Korea Centers for Disease Control & Prevention, Republic of Korea (http://www.kdca.go.kr).

**Competing interests:** The authors declare that there is no conflict of interest.

## Conclusions

Thus, the onset-to-door time of≤4.5 h is crucial for better clinical outcome, and lower NIHSS score, greater pre-stroke mRS score, female sex, unclear onset times, and ≤6 years of schooling were independent predictors of late arrival. Therefore, educating about the importance of early hospital arrival after acute ischemic stroke should be emphasized. More strategic efforts are needed to reduce the prehospital delay by understanding the predictors of late arrival.

## Introduction

Acute ischemic stroke is an emergency situation, and delayed treatment often leads to death or disability [1, 2]. The administration of intravenous tissue plasminogen activator (IV-tPA) within 4.5 h or performing mechanical thrombectomy within 24 h of stroke onset has been established as the most effective treatment, resulting in better clinical outcomes [3, 4]. However, since the eligibility for reperfusion therapy is highly dependent on stroke onset time [5], very few patients, including those who arrive at the hospital late or those with unclear onset time, miss the opportunity to receive such therapies [6, 7]. Early hospital arrival after acute ischemic stroke is also important for those who are eligible for reperfusion therapy because the sooner the hospital arrival, the better the efficacy of reperfusion therapy [8, 9]. However, despite the constant efforts to shorten the onset-to-door time of ischemic stroke, the proportion of eligible patients for reperfusion therapy has remained low, as many patients are still unable to receive this therapy [10–12].

Previous studies showed that early hospital arrival was a significant factor for better functional outcomes [13–15]. However, patients who received reperfusion therapy during the acute phase were included in most of these studies; therefore, better outcomes of early arrivers might be owing to reperfusion therapy. Additionally, since the reperfusion therapy is provided based on patients' last known well time, patients with unclear-onset stroke are less likely to receive reperfusion therapy [16]. Therefore, there is a lack of knowledge regarding whether arriving at a hospital early with unclear-onset stroke will benefit patients. That is, evidence regarding the association between early hospital arrival and better clinical outcomes in patients with ischemic stroke regardless of reperfusion therapy or the type of stroke onset time (clear- or unclear-onset strokes) is relatively limited.

To address this issue, we assessed the onset-to-door time in patients with acute ischemic stroke based on data from two large stroke centers in Korea. This study aimed to investigate whether early hospital arrival (onset-to-door time ≤4.5 h) is independently associated with favorable functional outcomes regardless of various potential factors, including receiving reperfusion therapy and the type of stroke onset time (clear- or unclear-onset strokes), that could affect the outcome. We also identified factors associated with late hospital arrival after acute ischemic stroke to offer a meaningful perspective to reduce pre-hospital delays.

## Methods

### Study institutions and population

Data for this two-center retrospective cohort study were obtained from the prospective stroke registries of Seoul National University Hospital (SNUH) and Chung-Ang University Hospital (CAUH). SNUH is a 1,750-bed university tertiary-referral hospital located in north-central Seoul. CAUH is also a university tertiary-referral hospital with 800 beds located in southwest

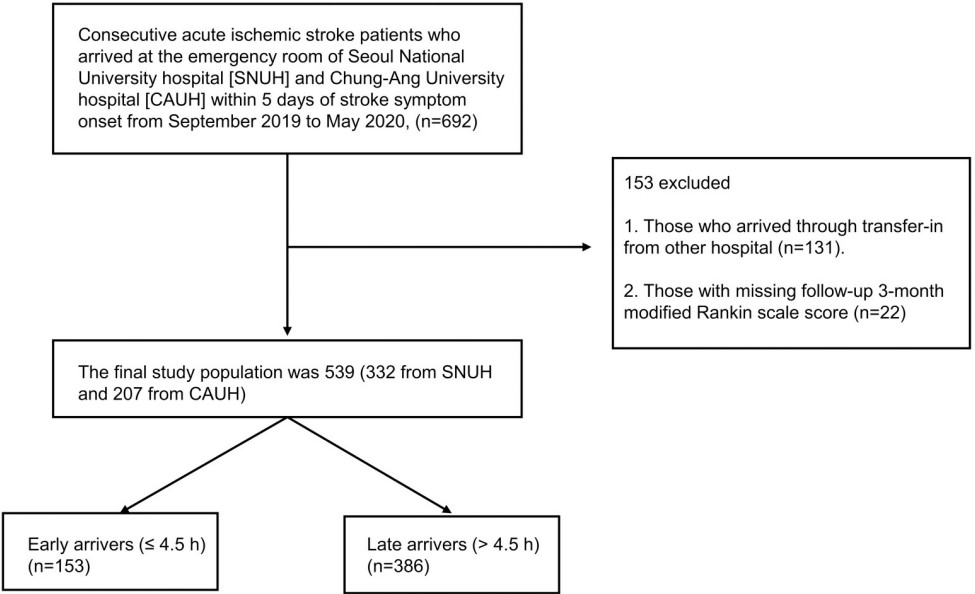

**Fig 1. Flowchart of the selection process of participants.** SNUH, Seoul National University Hospital; CAUH, Chung-Ang University Hospital.

Seoul, the capital of the Republic of Korea. Both stroke centers of the SNUH and CAUH fulfilled the major requirements of the comprehensive stroke center as defined by the Brain Attack Coalition [17]. A total of 692 consecutive patients with acute ischemic stroke who arrived at the emergency room of each hospital within five days of stroke symptom onset from September 2019 to May 2020 were included in the study. Patients who arrived via transfer from another hospital and whose three-month modified Rankin scale (mRS) scores were not available were excluded. In effect, the final study population was 539 patients (332 from SNUH and 207 from CAUH, Fig 1). Our study was approved by the institutional review boards (IRBs) of both the SNUH (IRB Number: 1009-062-332) and CAUH (IRB Number: 1912-005-359); the requirement for written consent was waived by the IRBs.

## Study variables and groups

The onset time of stroke, time of arrival at the hospital, sociodemographic data (sex, age, years of schooling), conventional vascular risk factors (history of hypertension, diabetes mellitus [DM], dyslipidemia, atrial fibrillation [AF], previous ischemic stroke or transient ischemic attack [TIA], coronary artery disease [CAD], and smoking status), medication history of antithrombotics (antiplatelet or anticoagulation agents), initial National Institute of Health Stroke Scale (NIHSS) score, pre-stroke mRS score, the status of receiving acute reperfusion therapy (IV-tPA administration or mechanical thrombectomy), and the details of hospital arrival, including daytime or nighttime arrival, of each patient were collected from the database of each hospital's stroke registry. Regarding the indicators of clinical outcomes of patients, the length of hospital stay, mRS scores at discharge, and mRS scores at three months after stroke were gathered. The onset time of stroke was defined as the time when the patient or bystander first noticed the stroke symptoms. For patients with uncertain onset time, the last known asymptomatic time was considered as the onset time. The hospital arrival time was categorized into daytime (07:01−21:00) and nighttime (21:00−07:00) according to the visit hours during the day. For smoking status, patients were categorized as either current smokers or

non-smokers (never or past smokers). Patients were dichotomized into two groups: the early arrival group (time from symptom onset to hospital arrival ≤4.5 h) and the late arrival group (time from symptom onset to hospital arrival >4.5 h). We selected 4.5 h as the cut-off point for categorizing early and late arrivers to reflect the current 4.5-h limit of IV-tPA administration [3].

## Predictors of delayed onset-to-door time and favorable outcomes

We included sex, age, educational status, the type of stroke onset time (clear- or unclear-onset strokes), medical histories (pre-stroke mRS score, comorbidities, previous stroke or TIA, previous use of antithrombotics, and smoking status), initial NIHSS score, and arrival time (daytime or nighttime) as potential factors that could affect the delayed onset-to-door time. Regarding the factors associated with favorable clinical outcome, age, educational status, the type of stroke onset time (clear- or unclear-onset strokes), medical histories (pre-stroke mRS score, comorbidities, previous stroke or TIA, previous use of antithrombotics and smoking status), initial NIHSS score, early hospital arrival (≤4.5 h), arrival time (daytime or nighttime), and receiving reperfusion therapy were included for the analysis. Both the pre-stroke mRS and initial NIHSS scores were included as continuous variables.

## Statistical analyses

The characteristics and outcome variables were presented as numbers, percentages, and means with corresponding standard deviations. We performed the independent sample *t*-test or Mann-Whitney U test for continuous variables, and chi square test or Fisher's exact test for categorical variables to compare the characteristics and outcome variables between the early and late arrival groups. With delayed arrival to a hospital (>4.5 h) as the dependent variable, univariate analysis was performed, followed by an adjusted multivariate logistic regression analysis for investigating the predictors of pre-hospital delay. Furthermore, univariate and multivariate logistic regression analyses were performed to examine the association between early hospital arrival and favorable outcomes. We included variables with p-values <0.1 in the univariate analysis for the multivariate analysis. All statistical analyses were performed using SPSS version 25 (IBM Corp., Armonk, NY, USA). A two-sided p-value of <0.05 was considered statistically significant.

# Results

## Baseline characteristics of patients

Among the 539 patients with ischemic stroke, 153 (28.4%) arrived at the hospital within 4.5 h of stroke onset, while 386 (71.6%) patients arrived at the hospital after 4.5 h. The mean age of patients was 68.3 ± 13.1 years. The mean onset-to-door time of all patients was 1761.31 ± 2217.91 min, whereas the mean onset-to-door time for the early and late arrival groups were 100.38 ± 68.72 min and 2,419.64 ± 2,311.10 min, respectively. Most patients (82.4%) arrived at the hospital during the daytime. Among all patients, 55.1% were men and 43.6% had unclear stroke onset. Regarding educational status, 33.9% of the patients had ≤6 years of schooling, 42.5% had 7–12 years of schooling, and the remaining 23.6% had >12 years of schooling. Regarding comorbidities, 56.0% of the patients had hypertension, 34.1% had DM, 46.9% had dyslipidemia, 19.3% had AF, and 14.3% had CAD. Those with a history of ischemic stroke or TIA accounted for 24.9% of the patients, and 43.6% had a history of antithrombotics use. Current smokers accounted for 29.3% of the patients, and 70.7% were either never smokers or past smokers. Regarding the severity of stroke and pre-stroke disability, the average initial NIHSS

score was 4.59 ± 5.65, and the average pre-stroke mRS score was 0.33 ± 0.96. Of the total ischemic stroke patients, 8.2% received intravenous thrombolysis, and 9.1% underwent mechanical thrombectomy. Regarding clinical outcomes, the average length of hospitalization was 11.21 ± 14.34 days, and the average NIHSS score, mRS score at discharge, and three-month mRS score were 3.39 ± 6.19, 1.88 ± 1.46, and 1.79 ± 1.48, respectively. Furthermore, seven patients died within three months since the event, and all of them were late arrivers. The early arrivers were more likely to be younger, men, more educated and have clear-onset stroke, lesser pre-stroke disability, greater severity of the stroke, and better mRS score at discharge and three-month mRS score. The differences in other variables between the early and late arrivers were not statistically significant (Table 1).

## Factors associated with delayed onset-to-door time

The results of the multivariate logistic regression analysis of factors associated with delayed onset-to-door time (>4.5 h) are presented in Table 2. Female sex (adjusted odds ratio [aOR]: 1.71, 95% confidence interval [CI]: 1.14–2.58) and a higher previous mRS score (aOR: 1.58, 95% CI: 1.18–2.13) were significantly associated with delayed onset-to-door time. However, clear-onset stroke (aOR: 0.35, 95% CI: 0.22–0.53) and a higher initial NIHSS score (aOR: 0.94, 95% CI: 0.90–0.97) showed a negative significant correlation with late hospital arrival. For educational status, low educational level (0–6 years of schooling) (aOR: 1.76, 95% CI: 1.03–3.00) was significantly associated with late hospital arrival than high educational level (>12 years); however, no significant association was found between intermediate (7–12 years) and high educational levels. Other factors, including age, comorbidities (hypertension, DM, AF, and CAD), smoking status, history of stroke or TIA, previous use of antithrombotics, and arrival at daytime or nighttime, were not significantly associated with delayed onset-to-door time. The multivariate model showed a good fit (Hosmer–Lemeshow test: P = 0.689).

## Factors associated with a favorable outcome (3-month mRS score: 0–2)

The results of the multivariate logistic regression analysis of the predictors of favorable outcome (three-month mRS: 0–2) are presented in Table 3. Overall, early hospital arrival within 4.5 h was the most prominent factor (aOR: 2.03, 95% CI: 1.04–3.96) that significantly correlated with a favorable three-month mRS score. Moreover, older age (aOR: 0.97, 95% CI: 0.95–0.99), higher previous mRS score (aOR: 0.52, 95% CI: 0.38–0.69), and initial NIHSS score (aOR: 0.77, 95% CI: 0.73–0.82) had a significant negative association with favorable outcomes. Other factors, including age, educational level, comorbidities (hypertension, DM, AF, and CAD), previous ischemic stroke or TIA, previous use of antithrombotics, acute thrombolysis therapy, and arrival at daytime or nighttime, had no significant correlation with a favorable three-month mRS. The multivariate model showed a good fit (Hosmer–Lemeshow test: p-value = 0.756).

## Discussion

Our study demonstrated that patients who arrived at the hospital within 4.5 h of stroke onset had significantly lower three-month mRS scores after the stroke event than those who arrived after 4.5 h. Furthermore, the results of the multivariate analysis demonstrated that less severe stroke and pre-stroke disability, younger age, and early hospital arrival within 4.5 h of stroke onset were significant predictors of favorable clinical outcome (three-month mRS score: 0–2). Among these significant factors, early hospital arrival was the most prominent determinant of better prognosis even after adjusting for patients' every other baseline characteristic. Previous studies have also confirmed that early hospital arrival within 1–6 h of stroke onset was an

**Table 1. Baseline characteristics of patients according to onset-to-door time.**

| Characteristics | All n (%) or mean ± SD | Onset-to-door time | | p-value |
|---|---|---|---|---|
| | | Early arrival group (0–4.5 h) n (%) or mean ± SD | Late arrival group (>4.5 h) n (%) or mean ± SD | |
| Total | 539 (100%) | 153 (100%) | 386 (100%) | |
| Socio-demographic factors | | | | |
| Sex | | | | 0.004 |
| Male | 297 (55.1%) | 99 (64.7%) | 198 (51.3%) | |
| Female | 242 (44.9%) | 54 (35.3%) | 188 (48.7%) | |
| Age (years) | 68.3±13.05 | 66.47±11.89 | 69.02±13.43 | 0.041 |
| <50 | 43 (8.0%) | 13 (8.5%) | 30 (7.8%) | |
| 50−59 | 80 (14.8%) | 26 (16.9%) | 54 (13.9%) | |
| 60−69 | 143 (26.5%) | 46 (30.1%) | 97 (25.1%) | |
| 70−79 | 168 (31.2%) | 49 (32.0%) | 119 (30.8%) | |
| ≥80 | 105 (19.5%) | 19 (12.4%) | 86 (22.3%) | |
| Years of schooling | | | | 0.023 |
| 0−6 years | 183 (33.9%) | 40 (21.9%) | 144 (37.3%) | |
| 7−12 years | 229 (42.5%) | 68 (44.4%) | 161 (41.7%) | |
| >12 years | 127 (23.6%) | 45 (29.4%) | 81 (21.0%) | |
| Type of stroke onset time | | | | 0.000 |
| Clear | 304 (56.4%) | 110 (71.9%) | 194 (50.3%) | |
| Unclear | 235 (43.6%) | 43 (28.1%) | 192 (49.7%) | |
| Medical history | | | | |
| Previous mRS score | 0.33±0.96 | 0.16±0.67 | 0.40±1.04 | 0.001 |
| Comorbidities | | | | |
| Hypertension | 377 (69.9%) | 109 (71.2%) | 268 (69.4%) | 0.813 |
| Diabetes mellitus | 184 (34.1%) | 53 (34.6%) | 131 (33.9%) | 0.759 |
| Dyslipidemia | 253 (46.9%) | 76 (49.6%) | 177 (45.8%) | 0.424 |
| Atrial fibrillation | 104 (19.3%) | 32 (20.9%) | 72 (18.7%) | 0.549 |
| Coronary artery disease | 77 (14.3%) | 27 (17.6%) | 50 (13%) | 0.185 |
| Previous ischemic stroke or TIA | | | | 0.318 |
| Yes | 134 (24.9%) | 42 (27.5%) | 92 (23.93%) | |
| No | 405 (75.1%) | 111 | 294 | |
| Previous use of antithrombotics[a] | | | | 0.475 |
| Yes | 235 (43.6%) | 63 (41.2%) | 172 (44.6%) | |
| No | 304 (56.4%) | 90 | 214 | |
| Smoking status | | | | 0.653 |
| Current smoker | 158 (29.3%) | 47 (30.7%) | 111 (28.8%) | |
| Non-smoker (never/past smoker) | 381 (70.7%) | 106 | 275 | |
| Initial NIHSS score | 4.59±5.65 | 5.55±6.70 | 4.22±5.14 | 0.028 |
| Acute reperfusion therapy | | | | |
| IV-tPA | 44 (8.2%) | 44 (28.8%) | 0 (0.0%) | 0.000 |
| IA thrombectomy | 49 (9.1%) | 29 (19.0%) | 20 (5.2%) | 0.000 |
| Outcomes | | | | |
| Length of hospital stay | 11.21±14.34 | 12.89±32.72 | 10.55±12.29 | 0.390 |
| NIHSS score at discharge | 3.39±6.19 | 2.78±4.41 | 3.63±6.76 | 0.149 |
| mRS score at discharge | 1.88±1.46 | 1.64±1.37 | 1.98±1.48 | 0.016 |
| mRS score at 3 months | 1.79±1.48 | 1.33±1.36 | 1.89±1.48 | 0.000 |
| Mortality at 3 months | 7 (1.30%) | 0 (0%) | 7 (0.02%) | 0.200 |

(*Continued*)

**Table 1.** (Continued)

| Characteristics | All n (%) or mean ± SD | Onset-to-door time | | p-value |
|---|---|---|---|---|
| | | Early arrival group (0–4.5 h) n (%) or mean ± SD | Late arrival group (>4.5 h) n (%) or mean ± SD | |
| Arrival time | | | | |
| Arrival at daytime (7 AM–9 PM) | 444 (82.4%) | 122 (79.7%) | 322 (83.4%) | 0.330 |
| Arrival at nighttime (9 PM–7 AM) | 95 (17.6%) | 31 (20.3%) | 64 (16.6%) | 0.314 |
| Mean onset-to-door time (mins) | 1761.31±2217.91 | 100.38±68.72 | 2419.64±2311.10 | 0.000 |

Abbreviations: SD, standard deviation; mRS, modified Rankin Scale; TIA, transient ischemia stroke; IV, intravenous; tPA tissue plasminogen activator; IA, intra-arterial; NIHSS, National Institute of Health Stroke Scale; AM, ante meridiem; PM, post meridiem.

[a]Antiplatelets or anticoagulation agents.

independent predictor for better prognosis regardless of patients' demographic factors, conventional vascular risk factors, or stroke severity [13–15]. However, these studies mostly credited the effect of reperfusion therapy for better outcomes in the early arrival group or did not distinguish patients who received IV-tPA or mechanical thrombectomy. In contrast, our study confirmed a favorable outcome of early arrivers even after adjusting for reperfusion therapy.

The findings of a recent multicenter Japanese study were also consistent with those of our study: early arrivers had a better clinical outcome regardless of reperfusion therapy [10]. In fact, our study has proven that early hospital arrival was associated with favorable clinical outcomes not only regardless of sociodemographic factors, conventional vascular risk factors,

**Table 2. Factors associated with delayed onset-to-door time (> 4.5 h).**

| Factors | Univariate analysis (n = 539) | | Multivariate analysis (n = 539) | |
|---|---|---|---|---|
| | Unadjusted OR (95% CI) | p-value | Adjusted OR[a] (95% CI) | p-value |
| Female sex | 1.74 (1.18–2.56) | 0.005 | 1.71 (1.14–2.58) | 0.010 |
| Age (years) | 1.02 (1.00–1.03) | 0.042 | 1.01 (0.99–1.03) | 0.267 |
| Education | | | | |
| >12 years | ref | | ref | |
| 7–12 years | 1.32 (0.83–2.09) | 0.245 | 1.18 (0.70–1.86) | 0.606 |
| 0–6 years | 2.00 (1.21–3.32) | 0.007 | 1.76 (1.03–3.00) | 0.017 |
| Clear onset | 0.40 (0.26–0.59) | 0.000 | 0.35 (0.22–0.53) | 0.000 |
| Previous mRS score | 1.44 (1.09–1.90) | 0.011 | 1.58 (1.18–2.13) | 0.003 |
| Hypertension | 0.95 (0.63–1.44) | 0.812 | - | - |
| Diabetes mellitus | 0.98 (0.65–1.44) | 0.877 | - | - |
| Dyslipidemia | 0.86 (0.59–1.25) | 0.423 | - | - |
| Atrial fibrillation | 0.87 (0.54–1.38) | 0.549 | - | - |
| Coronary artery disease | 0.69 (0.42–1.16) | 0.162 | - | - |
| Current smoker | 0.91 (0.61–1.37) | 0.652 | - | - |
| Previous ischemic stroke or TIA | 0.79 (0.52–1.21) | 0.273 | - | - |
| Previous use of antithrombotics[b] | 1.15 (0.79–1.68) | 0.475 | - | - |
| Initial NIHSS score | 0.96 (0.93–0.99) | 0.015 | 0.94 (0.90–0.97) | 0.000 |
| Arrival at daytime | 1.28 (0.79–2.06) | 0.313 | - | - |
| Arrival at nighttime | 0.77 (0.48–1.24) | 0.278 | - | - |

[a]Adjusted for sex, age, educational status, type of stroke onset time (clear- or unclear-onset strokes), previous mRS score, initial NIHSS score.

[b]Antiplatelets and anticoagulation agents.

Abbreviations: OR, odds ratio; CI, confidence interval; mRS, modified Rankin Scale; TIA, transient ischemic stroke; NIHSS, National Institute of Health Stroke Scale.

**Table 3. Factors associated with a favorable outcome (3-month mRS score: 0–2).**

| Factors | Univariate analysis (n = 539) | | Multivariate analysis (n = 539) | |
|---|---|---|---|---|
|  | OR (95% CI) | p-value | Adjusted OR[a] (95% CI) | p-value |
| Early arrival at hospital (≤4.5 h) | 1.62 (1.04–2.52) | 0.032 | 2.03 (1.04–3.96) | 0.039 |
| Age (years) | 0.95 (0.94–0.97) | 0.000 | 0.97 (0.95–0.99) | 0.000 |
| Education |  |  |  |  |
| >13 years | ref |  | - | - |
| 7–12 years | 0.75 (0.45–1.26) | 0.282 | - | - |
| 0–6 years | 0.74 (0.45–1.22) | 0.236 | - | - |
| Clear onset | 1.93 (1.32–2.82) | 0.001 | 1.41 (0.86–2.31) | 0.169 |
| Previous mRS score | 0.45 (0.35–0.57) | 0.000 | 0.52 (0.38–0.69) | 0.000 |
| Hypertension | 1.30 (0.85–1.99) | 0.221 | - | - |
| Diabetes mellitus | 1.05 (0.71–1.55) | 0.822 | - | - |
| Dyslipidemia | 1.28 (0.88–1.86) | 0.202 | - | - |
| Atrial fibrillation | 1.71 (1.09–2.68) | 0.020 | 1.01 (0.54–1.87) | 0.981 |
| Coronary artery disease | 1.27 (0.76–2.13) | 0.370 | - | - |
| Current smoker | 0.58 (0.37–0.90) | 0.016 | 1.48 (0.85–2.55) | 0.163 |
| Previous ischemic stroke or TIA | 0.68 (0.45–1.03) | 0.070 | 0.72 (0.42–1.24) | 0.236 |
| Previous use of antithrombotics[b] | 0.75 (0.52–1.1) | 0.136 | - | - |
| Acute thrombolysis therapy (IV-tPA or IAT) | 0.54 (0.32–0.91) | 0.020 | 2.09 (0.87–5.03) | 0.101 |
| Initial NIHSS score | 0.79 (0.75–0.83) | 0.000 | 0.77 (0.73–0.82) | 0.000 |
| Arrival at daytime | 0.95 (0.58–1.56) | 0.843 | - | - |
| Arrival at nighttime | 1.03 (0.63–1.70) | 0.898 | - | - |

[a]Adjusted for arrival time at hospital, age, type of stroke onset time (clear- or unclear-onset strokes), previous mRS score, history of atrial fibrillation, smoking status, previous ischemic stroke or TIA, acute thrombolysis therapy, and initial NIHSS score.

[b]Antiplatelets and anticoagulation agents.

Abbreviations: OR, odds ratio; CI, confidence interval; mRS, modified Rankin Scale; TIA, transient ischemic stroke; IV, intravenous; tPA, tissue plasminogen activator; IAT, intra-arterial thrombectomy; NIHSS, National Institute of Health Stroke Scale.

severity of a stroke, previous stroke or TIA, previous use of antithrombotics, pre-stroke disability, and receiving reperfusion therapies, but also regardless of the type of stroke onset (clear- or unclear-onset stroke). Thus, the importance of reporting to the hospital as soon as possible after recognizing stroke symptoms cannot be emphasized more in acute ischemic stroke patients.

Regarding the factors associated with delayed hospital arrival after ischemic stroke (>4.5 h), an additional multivariate analysis identified that lesser severity of the stroke, greater pre-stroke disability, female sex, unclear onset times, and low educational level with ≤6 years of schooling were significant factors that influenced late hospital arrival in acute ischemic stroke patients. The severity of the initial stroke significantly correlated with the onset-to-door time in our study because patients with lower NIHSS scores arrived notably later than those with higher scores. This may be because the greater severity of the stroke would have likely had a greater impact on the perceived urgency of patients or bystanders to take quick actions to visit the hospital for emergency care. Rosamond et al. reported that an increased sense of urgency by stroke patients for their symptoms positively influenced early hospital arrival [18]. Lesser severity has also been well established as a significant factor for late hospital arrival in several previous studies using various measures for severity [19–26].

In contrast, greater pre-stroke disability, as estimated with the mRS, was markedly associated with late hospital arrival in our study, which was contrary to that of previous studies

which reported that greater pre-stroke disability was associated with increased emergency medical service (EMS) use [27], or it was associated with an early hospital arrival [28]. The likelihood of increased physical and intellectual impairment in patients with a greater pre-stroke disability may explain our findings. Although patients with greater disability receive more EMSs, limited mobility or decreased cognitive function and other factors related to pre-stroke disability could have caused considerable delay in calling for EMSs, which contributed to the overall prehospital delay. A stroke registry-based study conducted by Madsen et al. also presented a similar trend as ours and interpreted their findings on this basis [29].

Interestingly, there was an association between sex difference and hospital arrival time in our study because women tended to arrive at the hospital later than men. According to Bergu-lund et al., women are generally older, have more impaired activities of daily living, and are more likely to live alone than men during stroke onset [30]. In fact, approximately 71% of the elderly living alone in Korea are women [31]. Additionally, when ischemic stroke patients are in contact with EMSs, the caller is usually their relative or friend rather than the patients them-selves [32, 33]; hence, those living alone would obviously have more difficulty seeking emer-gent help. Previous studies have confirmed that living alone during stroke onset is a noticeable risk factor for late hospital arrival in acute ischemic stroke patients [33, 34]. As previously reported, female sex was also a risk factor for late hospital arrival in patients with ischemic stroke because women are more likely to be living alone than men [26, 35].

Another notable finding of our study was that unclear onset time of stroke was a significant predictor of late hospital arrival. To the best of our knowledge, our study is the first to examine the relationship between the type of stroke onset time (clear- or unclear-onset strokes) and onset-to-door time in acute ischemic stroke. Unclear-onset stroke occurs in patients who were asleep and who woke up with the presence of stroke symptoms (wake-up stroke) or patients who could not state the onset time owing to loss of consciousness but with no available wit-nesses [36], thus resulting in late hospital arrival. Our results identified that poorly educated patients with ≤6 years of schooling report to the hospital later than well-educated individuals with >12 years of schooling. Well-educated patients are more likely to be aware of the symp-toms and signs that are indicative of acute ischemic stroke, which may have led them to promptly report to the hospital for urgent evaluation and management. This was consistent with the findings of previous studies that included patients with acute ischemic stroke or myo-cardial infarction [37, 38]. Thus, public education regarding the red flag symptoms of stroke and the importance of reporting early to the hospital should be implemented, especially to those with ≤6 years of schooling.

The strength of our study is that we analyzed data from two university tertiary hospitals with large stroke centers located in different areas of Seoul with different characteristics. Both the hospitals provided professional stroke care by experts according to the standard guidelines. Furthermore, we confirmed that early arrival in the hospital was an independent factor for a better clinical outcome regardless of potential confounders, including receiving reperfusion therapy and the type of stroke onset time (clear- or unclear-onset strokes). We also identified the predictors of late arrival, which provides an important reference for reducing pre-hospital delays.

However, our study has some limitations. First, other factors that may affect the onset-to-door time, such as distance from the place of stroke onset to the hospital and the type of trans-port system that patients used to report to the hospital, including EMS use, could not be included in the analysis owing to unavailability of data. Second, the onset-to-door time might have been overestimated in some patients, especially in those with unclear onset time, includ-ing those who had a wake-up stroke, since we defined the patients' last neurologically normal time as the symptom onset time when the exact symptom onset time was unknown. Lastly, the

patient cohort included in our study was mostly homogenous in terms of race and culture. Therefore, the findings of this study cannot be generalized, and further studies involving multiethnic and multicultural patients are needed to validate our findings.

## Conclusions

In summary, early hospital arrival within 4.5 h after stroke onset was significantly associated with better clinical outcomes at three months after the event regardless of various baseline characteristics of patients, including the severity of the stroke, pre-stroke disability, receiving reperfusion therapies, and the type of stroke onset time (clear- or unclear-onset strokes). Furthermore, less severe stroke, greater pre-stroke disability, female sex, unclear onset times, and ≤6 years of schooling were independent factors that significantly affected late hospital arrival. Therefore, educating patients about the importance of early hospital arrival after acute ischemic stroke should be emphasized, and more strategic efforts are needed to reduce the prehospital delay by understanding the factors associated with a late arrival.

## Supporting information

**S1 Dataset. Minimal dataset.**
(XLSX)

## Author Contributions

**Conceptualization:** Eung-Joon Lee, Seung Jae Kim, Hae-Bong Jeong.

**Data curation:** Eung-Joon Lee, Jeonghoon Bae, Oh Deog Kwon, Yongsung Kim.

**Formal analysis:** Eung-Joon Lee, Eun Ji Lee.

**Software:** Eun Ji Lee, Oh Deog Kwon.

**Supervision:** Han-Yeong Jeong.

**Writing – original draft:** Eung-Joon Lee, Seung Jae Kim.

**Writing – review & editing:** Hae-Bong Jeong.

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
