## [Decision Letter · Decision Letter 0]

25 Jan 2021

PONE-D-20-39466

Impact of Onset-to-Door Time on Outcomes and Factors Associated with Late Arrival in Acute Ischemic Stroke Patients

PLOS ONE

Dear Dr. Jeong,

Thank you for submitting your manuscript to PLOS ONE. After careful consideration, we feel that it has merit but does not fully meet PLOS ONE’s publication criteria as it currently stands. Therefore, we invite you to submit a revised version of the manuscript that addresses the points raised during the review process.

The manuscript is in general well written, but I encourage the authors to check for English Editing Services as there are some minor grammatical mistakes throughout the manuscript. Some minor issues from the methodological point of view, mainly in terms of covariates included in the multivariate analysis: did you perform a goodness of fit test when (such as Hosmer-Lemeshow) for correction of the final model? Also, did you pre-establish a p-value to include variables in the multivariate model, according to your pre-defined literature research?

Find attached also some suggestions from the 2 reviewers,

We look forward to receiving your revised manuscript.

Kind regards,

Miguel A. Barboza, MD, MSc

Academic Editor

PLOS ONE

Journal Requirements:

Reviewers' comments:

Reviewer's Responses to Questions

**Comments to the Author**

1. Is the manuscript technically sound, and do the data support the conclusions?

Reviewer #1: Yes

Reviewer #2: Yes

2. Has the statistical analysis been performed appropriately and rigorously? 

Reviewer #1: Yes

Reviewer #2: Yes

3. Have the authors made all data underlying the findings in their manuscript fully available?

Reviewer #1: Yes

Reviewer #2: Yes

4. Is the manuscript presented in an intelligible fashion and written in standard English?

Reviewer #1: Yes

Reviewer #2: Yes

5. Review Comments to the Author

Reviewer #1: We thank the authors for submitting their manuscript to the PLOS ONE Journal. The authors present a paper analysing clinical outcomes in acute ischemic stroke in a population with early vrs late hospital arrival independently from reperfusion therapy.

In spite of being a topic that has been published, at least partially, in the literature it is of significance and interest to our readers. The overall organization of the article is excellent. I do not feel any section should be expanded or shortened. The writing is clear, concise and all the sections reflect its main point of view. My overall impression is good.

The introduction is short but strong, providing an adequate background review of the literature related to the topic of the study. The main point of the article and hypothesis are clearly stated.

The abstract accurately reflects the paper.

The methods provided are valid for the question asked and are clearly presented so that this work could be replicated correctly. The statistical analysis was well done for the variables used. The sample size is ok and most of the variables analysed reach statistical significance. That said, bigger sample could always improve the significance and provide stronger evidence and validity of the study; especially for some of the vascular risk factors, such as dyslipidemia and diabetes, among others.

The authors´ results and data are clearly summarized and provided in tables and figures, which are necessary for the article. I do not see information needlessly repeated.

The discussion and conclusions were good, providing a powerful assessment of the results. Limitations were identified and explained.

The only figure was of moderate quality and could be improved. Both figure and tables were clearly labeled and titled.

The citations were pertinent and current. They all support assertions of fact that were not addressed by the data presented on the paper.

Reviewer #2: Dear Dr. HaeBong Jeong,

Thank you for permitting us to read your interesting manuscript.

This is a retrospective study of a prospective cohort, that investigated the impact of onset-to-door time on outcomes and predictors of pre-hospital delay after ischemic, including 539 patients from Seoul population, between 2019–2020. The conclusion was that early hospital arrival was significantly associated with favorable outcomes regardless of various confounders (reperfusion therapy, NIHSS, etc). It was a great effort, congratulations. Please consider the following comments.

1.- It should be important to mention in the methodology what is called “greater or lower pre-stroke disability”, in most cases we used a mRs 0-2 and 3-6, but this data should be clear in the study. The same case for NIHSS. I suggest that this difference also becomes noticeable in the tables, because according to this, it is the interpretation of the OR in them.

2.- In the discussion, I consider that it is more useful for the reader to know which mRS range predicts better outcomes (example 0-2, 3-6), more than the average mRS itself (example 0.33).

3.- The proposal made at the conclusion of the abstract is excellent, however, here the question of the title should also be answered. I suggest adding a statement that the onset to door time is important for prognosis and, briefly mentioning the factors involved in the arrival hospital delay, followed by your excellent education suggestion.

Thanks,

6. PLOS authors have the option to publish the peer review history of their article (what does this mean?). If published, this will include your full peer review and any attached files.

Reviewer #1: **Yes: **Karla Alejandra Mora Rodriguez. Neurology Attending Physician. Epilepsy and Clinical Neurophysiology Subspecialty.

Reviewer #2: No

---

## [Author Response · Author response to Decision Letter 0]

30 Jan 2021

Response to the Editor’s requests:

1. The manuscript is in general well written, but I encourage the authors to check for English Editing Services as there are some minor grammatical mistakes throughout the manuscript.

First of all, we apologize you for causing confusion. We have revised grammatical mistakes through English Editing Services as you recommended. Thank you for your thoughtful advice.

Please find the attached revised manuscript. 

2. Some minor issues from the methodological point of view, mainly in terms of covariates included in the multivariate analysis: did you perform a goodness of fit test when (such as Hosmer-Lemeshow) for correction of the final model? Also, did you pre-establish a p-value to include variables in the multivariate model, according to your pre-defined literature research?

We would like to thank the editor for pointing this issue which has substantially helped us improve the methodology and overall quality of our study findings. Regarding the selection of covariates for the multivariate analyses of this study, we did perform Hosmer-Lemeshow goodness of fit test for each confounding variables for all of multivariate logistic regression analyses and the results demonstrated that there were no differences between observed and expected values as below, indicating that our models were suitable for regression analyses. 

The results of Hosmer-Lemeshow test

1. Multivariate logistic regression analyses for factors associated with favorable outcome (3-month mRS score 0-2)

χ²: 5.016, p value: 0.756

2. Multivariate logistic regression analyses for factors associated with delayed onset-to-door time (>4.5h)

χ²: 5.628, p value: 0.689

In terms of pre-establishing a p-value to include variables in the multivariate model, we actually did not perform univariate analyses before conducting multivariate analyses. Thus we have performed univariate analyses and re-conducted multivariate analyses by including variables with p values < 0.1 in the univariate analyses to analyze adjusted odds ratios and 95% confidence intervals in accordance with your thoughtful advice. The overall tendency of new results was consistent with the initial results. We have revised our tables and “Results” section of our manuscript with newly attained results. Please find the attached revised manuscript. 

We also revised the “Methods” section of our manuscript as below (newly added or revised contents are highlighted in yellow). 

Predictors of delayed onset-to-door time and favorable outcomes

We included sex, age, educational status, the type of stroke onset time (clear- and unclear-onset strokes), medical histories (pre-stroke mRS score, comorbidities, previous stroke or TIA, previous use of antithrombotics, and smoking status), initial NIHSS score, and arrival time (daytime or nighttime) as potential factors that could affect the delayed onset-to-door time. Regarding the factors associated with favorable clinical outcome, age, educational status, the type of stroke onset time (clear- or unclear-onset strokes), medical histories (pre-stroke mRS score, comorbidities, previous stroke or TIA, previous use of antithrombotics and smoking status), initial NIHSS score, early hospital arrival (≤4.5 h), arrival time (daytime or nighttime), and receiving reperfusion therapy were included for the analysis. Both the pre-stroke mRS and initial NIHSS scores were included as continuous variables. 

Statistical analyses

The characteristics and outcome variables were presented as numbers, percentages, and means with corresponding standard deviations. We performed the independent sample t-test or Mann-Whitney U test for continuous variables, and chi square test or Fisher's exact test for categorical variables to compare the characteristics and outcome variables between the early and late arrival groups. With delayed arrival to a hospital (>4.5 h) as the dependent variable, univariate analysis was performed, followed by an adjusted multivariate logistic regression analysis for investigating the predictors of pre-hospital delay. Furthermore, univariate and multivariate logistic regression analyses were performed to examine the association between early hospital arrival and favorable outcomes. We included variables with p-values <0.1 in the univariate analysis for the multivariate analysis. All statistical analyses were performed using SPSS version 25 (IBM Corp., Armonk, NY, USA). A two-sided p-value of <0.05 was considered statistically significant.

Response to the Reviewer #1’s comments:

1. We thank the authors for submitting their manuscript to the PLOS ONE Journal. The authors present a paper analysing clinical outcomes in acute ischemic stroke in a population with early vrs late hospital arrival independently from reperfusion therapy.

In spite of being a topic that has been published, at least partially, in the literature it is of significance and interest to our readers. The overall organization of the article is excellent. I do not feel any section should be expanded or shortened. The writing is clear, concise and all the sections reflect its main point of view. My overall impression is good.

The introduction is short but strong, providing an adequate background review of the literature related to the topic of the study. The main point of the article and hypothesis are clearly stated.

The abstract accurately reflects the paper.

The methods provided are valid for the question asked and are clearly presented so that this work could be replicated correctly. The statistical analysis was well done for the variables used. The sample size is ok and most of the variables analysed reach statistical significance. That said, bigger sample could always improve the significance and provide stronger evidence and validity of the study; especially for some of the vascular risk factors, such as dyslipidemia and diabetes, among others.

The authors´ results and data are clearly summarized and provided in tables and figures, which are necessary for the article. I do not see information needlessly repeated.

The discussion and conclusions were good, providing a powerful assessment of the results. Limitations were identified and explained.

We would like to thank the reviewer for such a kind words on our paper. We greatly appreciate all your valuable comments. 

2. The only figure was of moderate quality and could be improved. Both figure and tables were clearly labeled and titled.

The citations were pertinent and current. They all support assertions of fact that were not addressed by the data presented on the paper.

We have revised the content and quality of our figure for better understanding of our selection process of study participants and to meet the publication standard for your journal. Please find the attached revised figure. Thank you for your thoughtful advice. 

Response to Reviewer #2’s comments: 

1. It should be important to mention in the methodology what is called “greater or lower pre-stroke disability”, in most cases we used a mRs 0-2 and 3-6, but this data should be clear in the study. The same case for NIHSS. I suggest that this difference also becomes noticeable in the tables, because according to this, it is the interpretation of the OR in them.

We would like to thank the reviewer for this precious comment, which points out important aspect of our study. We agree with you that in most cases pre-stroke mRS 0-2 is generally considered as lower pre-stroke disability whereas mRS 3-6 is regarded as greater pre-stroke disability. However, both the pre-stroke mRS and initial NIHSS scores were included as continuous variables rather than binary variables in all of our multivariable analyses. Thus, our results can be interpreted as the lower the initial NHISS score or the greater the pre-stroke mRS score the more delayed onset-to-door time for acute ischemic stroke patients. We have clarified this issue by adding a sentence in the “Methods” section (line 134 of page 7) of our study as below. 

Both the pre-stroke mRS and initial NHISS scores were included as continuous variables. 

2. In the discussion, I consider that it is more useful for the reader to know which mRS range predicts better outcomes (example 0-2, 3-6), more than the average mRS itself (example 0.33).

 We appreciate the reviewer for this excellent comment which identifies essential perspective of our study. We agree with you that presenting mRS score with certain range when mentioning the effect of pre-stroke mRS score on the favorable outcome could be helpful for readers. However, as we mentioned in the previous question, pre-stroke mRS scores were included as continuous variables with average scores rather than binary variable with specific cut-off point. We believe our finding could be also meaningful since it indicates that the higher pre-stroke mRS predicts better clinical outcome regardless of good (mRS 0-2)/bad (mRS 3-6) status of pre-stroke mRS score. 

3. The proposal made at the conclusion of the abstract is excellent, however, here the question of the title should also be answered. I suggest adding a statement that the onset to door time is important for prognosis and, briefly mentioning the factors involved in the arrival hospital delay, followed by your excellent education suggestion.

First of all, we would like to thank the reviewer for the positive comment on the conclusion part of our abstract. We would appreciate the reviewer’s thoughtful suggestion to add a statement that the onset to door time is important for prognosis and mentioning the identified factors influencing the pre-hospital delay as it has helped us to strengthen our conclusion. We have revised the conclusion of abstract (line 44 of page 2) as below to reflect your suggestion (newly added or revised contents are highlighted in yellow).

The onset-to-door time≤4.5h is crucial for better clinical outcome and less severe stroke, greater pre-stroke disability, female sex, unclear onset times, and ≤6 years of schooling were independent predictors of late arrival. Thus, educating about the importance of early hospital arrival after acute ischemic stroke should be emphasized. More strategic efforts are needed to reduce the pre-hospital delay by understanding the predictors of late arrival.

---

## [Decision Letter · Decision Letter 1]

15 Feb 2021

Impact of onset-to-door time on outcomes and factors associated with late hospital arrival in patients with acute ischemic stroke

PONE-D-20-39466R1

Dear Dr. Jeong,

We’re pleased to inform you that your manuscript has been judged scientifically suitable for publication and will be formally accepted for publication once it meets all outstanding technical requirements.

Kind regards,

Miguel A. Barboza, MD, MSc

Academic Editor

PLOS ONE

Additional Editor Comments (optional):

Reviewers' comments:

Reviewer's Responses to Questions

**Comments to the Author**

1. If the authors have adequately addressed your comments raised in a previous round of review and you feel that this manuscript is now acceptable for publication, you may indicate that here to bypass the “Comments to the Author” section, enter your conflict of interest statement in the “Confidential to Editor” section, and submit your "Accept" recommendation.

Reviewer #1: All comments have been addressed

Reviewer #2: All comments have been addressed

2. Is the manuscript technically sound, and do the data support the conclusions?

Reviewer #1: Yes

Reviewer #2: Yes

3. Has the statistical analysis been performed appropriately and rigorously? 

Reviewer #1: Yes

Reviewer #2: Yes

4. Have the authors made all data underlying the findings in their manuscript fully available?

Reviewer #1: Yes

Reviewer #2: Yes

5. Is the manuscript presented in an intelligible fashion and written in standard English?

Reviewer #1: Yes

Reviewer #2: Yes

6. Review Comments to the Author

Reviewer #1: I currently have no comments for the authors on this re-review. On my previous comment I mentioned my overall impression was excellent and my only suggestion was to improve their one figure presented.

Reviewer #2: Dear Corresponding Author,

Thank you for taking the comments into account and making the pertinent changes.

I have no further comments, congratulations for this great effort.

7. PLOS authors have the option to publish the peer review history of their article (what does this mean?). If published, this will include your full peer review and any attached files.

Reviewer #1: **Yes: **Karla A. Mora R. MD. Neurology Attending Physician, Fellowship in Epilepsy

Reviewer #2: **Yes: **Vanessa Cano-Nigenda

---

## [Editor Report · Acceptance letter]

1 Mar 2021

PONE-D-20-39466R1 

Impact of onset-to-door time on outcomes and factors associated with late hospital arrival in patients with acute ischemic stroke 

Dear Dr. Jeong:

I'm pleased to inform you that your manuscript has been deemed suitable for publication in PLOS ONE. Congratulations! Your manuscript is now with our production department. 

Kind regards, 

on behalf of

Dr. Miguel A. Barboza 

Academic Editor

PLOS ONE